# Sequence to Multi-Sequence Learning via Conditional Chain Mapping for Mixture Signals

**Jing Shi**[1,2,*], **Xuankai Chang**[1,*], **Pengcheng Guo**[1,3,*], **Shinji Watanabe**[1†], **Yusuke Fujita**[4],
**Jiaming Xu**[2], **Bo Xu**[2], **Lei Xie**[3]

[1]Center for Language and Speech Processing, Johns Hopkins University, U.S.A
[2]Institute of Automation, Chinese Academy of Sciences (CASIA), Beijing, China
[3]ASLP@NPU, School of Computer Science, Northwestern Polytechnical University, Xi'an, China
[4]Hitachi, Ltd. Research & Development Group, Japan
`shijing2014@ia.ac.cn`, `xchang14@jhu.edu`, `pcguo@nwpu-aslp.org`,
`shinjiw@ieee.org` ✉ , `yusuke.fujita.su@hitachi.com`

## Abstract

Neural sequence-to-sequence models are well established for applications which can be cast as mapping a single input sequence into a single output sequence. In this work, we focus on one-to-many sequence transduction problems, such as extracting multiple sequential sources from a mixture sequence. We extend the standard sequence-to-sequence model to a conditional multi-sequence model, which explicitly models the relevance between multiple output sequences with the probabilistic chain rule. Based on this extension, our model can conditionally infer output sequences one-by-one by making use of both input and previously-estimated contextual output sequences. This model additionally has a simple and efficient stop criterion for the end of the transduction, making it able to infer the variable number of output sequences. We take speech data as a primary test field to evaluate our methods since the observed speech data is often composed of multiple sources due to the nature of the superposition principle of sound waves. Experiments on several different tasks including speech separation and multi-speaker speech recognition show that our conditional multi-sequence models lead to consistent improvements over the conventional non-conditional models.

## 1 Introduction

Many machine learning tasks can be formulated as a sequence transduction problem, where a system provides an output sequence given the corresponding input sequence. Examples of such tasks include machine translation, which maps text from one language to another, automatic speech recognition (ASR), which receives a speech waveform and produces a transcription, and video captioning, which generates the descriptions of given video scenes. In recent years, the development of neural sequence-to-sequence (seq2seq) models [9, 40] with attention mechanisms has led to significant progress in such tasks [2, 51, 44, 48, 10, 5].

In reality, the observed data contains various entangled components, making the one-to-many sequence transduction for mixture signals a common problem in machine learning [37, 18, 32]. This problem often happens in audio and speech processing due to the sequential properties and superposition principle of sound waves. For example, given the overlapped speech signal, speech separation is a problem of extracting individual speech sources, and multi-speaker speech recognition is a problem

---

[*]equal contribution
[†]corresponding author

of decoding transcriptions of individual speakers. This type of problem is called the cocktail party problem [8, 3]. The existing methods to tackle this common sequence-to-multi-sequence problem can be roughly divided into two categories according to the correlation strength of multiple output sequences: serial mapping and parallel mapping. Serial mapping aims to learn the mappings through

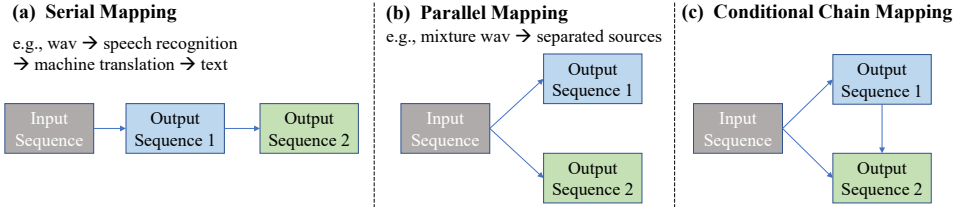

Figure 1: Sequence-to-multi-sequence mapping approaches. (a) and (b) shows the serial mapping and parallel mapping used by existing methods respectively, and (c) refers to our Conditional Chain mapping strategy.

a forward pipeline, as shown in Figure 1(a). With the serial form, the output sequence of the first seq2seq model is fed into the following seq2seq model to output another sequence. This method is quite common when the logic and relationship between different output sequences are straightforward. For example, a cross-lingual speech translation system contains two components: speech recognition and machine translation. Serial mapping first recognizes the speech into the source language text and then use another model to translate it into the target language text. However, serial mapping methods usually suffer from some drawbacks. First, many of them need to be trained separately for different components, without taking advantage of the raw input information in the latter components. And the error accumulation through the pipeline will make the system suboptimal. The other category is parallel mapping, as shown in Figure 1(b), which simultaneously outputs multiple sequences. This method is often used when the outputs are from the same domain. Speech separation and multi-speaker ASR are typical examples following this paradigm. Similar to serial mapping, parallel mapping could not effectively model the inherent relationship that exists between different outputs, and usually assumes the number of the output sequence is fixed (e.g., the fixed number of speakers in speech separation tasks), which limits its application scenarios.

In this paper, we propose a new unified framework aiming at the sequence-to-multi-sequence (seq2Mseq) transduction task, which can address the disadvantages of both the serial mapping and parallel mapping methods. For clarity, we refer to our methods as *Conditional Chain* (Cond-Chain) model, combining both the serial mapping and parallel mapping with the probabilistic chain rule. Simultaneous modeling for these two methods not only makes the framework more flexible but also encourages the model to automatically learn the efficient relationship between multiple outputs.

To instantiate the idea, as shown in Figure 1(c), we assume that the input sequence $O$ can be mapped into $N$ different sequences $s_i, i \in \{1, .., N\}$. We take sequence $O$ as the primary input for every output sequence. Meanwhile, the outputs will be generated one-by-one with the previous output sequence as a conditional input. We consider that the multiple outputs from the same input have some relevance at the information level. By combining both the serial and parallel connection, our model learns the mapping from the input to each output sequence as well as the relationship between the output sequences.

In this paper, we introduce the general framework in Section 2, and present a specific implementation for the tasks of speech separation and recognition in Section 3. We discuss some related work in Section 4 and describe our experiments in Section 5, and finally conclude in Section 6. Our source code and Supplementary Material could be available on our webpage: `https://demotoshow.github.io/`.

## 2   General framework

We assume that the input sequence $O \in \mathcal{O}$ with length of $T$ can be mapped into $N$ different sequences $s_i, i \in \{1, .., N\}$, where the output index $i$ represents a particular domain $\mathcal{D}_i$. All the output sequences form a set $\mathbf{S} = \{s_i \mid i \in \{1, ..., N\}\}$. The basic formulation of our strategy is to estimate the joint probability of multiple output sequences, i.e., $p(\mathbf{S}|O)$. The joint probability is factorized into the

product of conditional probabilities by using the probabilistic chain rule with/without the conditional independence assumption (denoted by -), as follows:

$$p(\mathbf{S}|O) = \begin{cases} p(s_1|O) \prod_{i=2}^{N} p(s_i|s_{i-1}, \cancel{O, s_{i-2}, ..., s_{\mathrm{T}}}) & \text{serial mapping} \\ \prod_{i=1}^{N} p(s_i|O, \cancel{s_{i-1}, ..., s_{\mathrm{T}}}) & \text{parallel mapping} \\ \prod_{i=1}^{N} p(s_i|O, s_{i-1}, ..., s_1) & \text{conditional chain mapping} \end{cases} \tag{1}$$

where, we also present the formulation of serial mapping and parallel mapping for comparison. As shown in Eq. 1, the serial mapping methods adopt a fixed order and the conditional distributions are only constructed with sequence from the previous step, i.e., $p(\mathbf{S}|O) = p(s_1|O) \prod_{i=2}^{N} p(s_i|s_{i-1})$. As a contrast, parallel mapping simplifies the joint distribution by the conditional independence assumption, which means all the output sequences are only conditioned on the raw input, i.e., $p(\mathbf{S}|O) = \prod p(s_i|O)$. For our conditional chain mapping, we manage to explicitly model the inherent relevance from the data, even if it seems very independent intuitively. To achieve this, we depart from the conditional independence assumption in parallel mapping or the Markov assumption in serial mapping. Instead, with the probabilistic chain rule, our method models the joint distribution of output sequences over an input sequence $O$ as a product of conditional distributions. We can also apply the same methodology to the non-probabilistic regression output (e.g., speech separation).

In our model, each distribution $p(s_i|O, s_{i-1}, ..., s_1)$ in Eq. 1 is represented with a conditional encoder-decoder structure. Different from the conventional one-to-one sequence transduction for learning the mapping $\mathcal{O} \mapsto \mathcal{D}_i$, additional module in our model preserves the information from previous target sequences and takes it as a condition for the following targets. This process is formulated as follows:

$$\mathbf{E}_i = \text{Encoder}_i(O) \in \mathbb{R}^{D_i^E \times T_i^E}, \tag{2}$$

$$\mathbf{H}_i = \text{CondChain}(\mathbf{E}_i, \hat{\mathbf{s}}_{i-1}) \in \mathbb{R}^{D_i^H \times T^H}, \tag{3}$$

$$\hat{\mathbf{s}}_i = \text{Decoder}_i(\mathbf{H}_i) \in \mathcal{D}_i^{T_i}, \tag{4}$$

where, all the $D_i$ symbols are the number of dimensions for the features, and $T_i^E, T_i^H, T_i$ represent the size of temporal dimension. In the above equations, $\text{Encoder}_i$ and $\text{Decoder}_i$ refer to the specific networks designed for learning the mapping for the reference sequence $s_i$. Note that the $\text{Encoder}_i$ and $\text{Decoder}_i$ here may also consist of linear layers, attention mechanism or other neural networks besides the standard RNN layer, so the lengths of the hidden embeddings $\mathbf{E}_i$ and the estimation sequence $\hat{\mathbf{s}}_i$ may vary from the input, i.e., $T_i^E, T_i^H, T_i$ may not equal the $T$. For the $i$-th output, the $\mathbf{E}_i$ in Encoder gets a length of $T_i^E$ while the $\hat{\mathbf{s}}_i$ should get the same length with the reference $s_i \in \mathcal{D}_i^{T_i}$, where $T_i$ is the length of the sequence $s_i$ from domain $\mathcal{D}_i$. Different from the conventional seq2seq model, we utilize a conditional chain ($\text{CondChain}$ in Eq. 3) to store the information from the previous sequences and regard them as conditions. This conditional chain is analogous to the design of memory cell in the LSTM model and the key component to realize Figure 1(c). Similarly, the conditional chain in Eq. 3 does not serve a specific target domain alone, it models some unified information for multi-sequence outputs. In other words, the encoder-decoder is specialized for each target sequence, but the conditional chain is shared by all the transduction steps $i$.

For most situations, when the logic and relationship between different output sequences is straightforward, we could set a fixed ordering of the outputted sequence, like the cross-lingual speech translation showed in Figure 1(a). Differently, for the outputs from the same domain, i.e., $\mathcal{D}_i = \mathcal{D}_j, i \neq j$, the Encoder and Decoder for each step could be shared with the same architecture and parameters, which yields less model parameters and better efficiency for training.

## 3 Implementation for speech processing

This section describes our implementation of the proposed conditional chain model by using specific multi-speaker speech separation / recognition tasks as examples. Both of them are typical examples of seq2Mseq tasks with input from mixture signals.

### 3.1 Basic model

Multi-speaker speech separation / recognition aims at isolating individual speaker's voices from a recording with overlapped speech. Figure 2 shows the network structure under our conditional chain

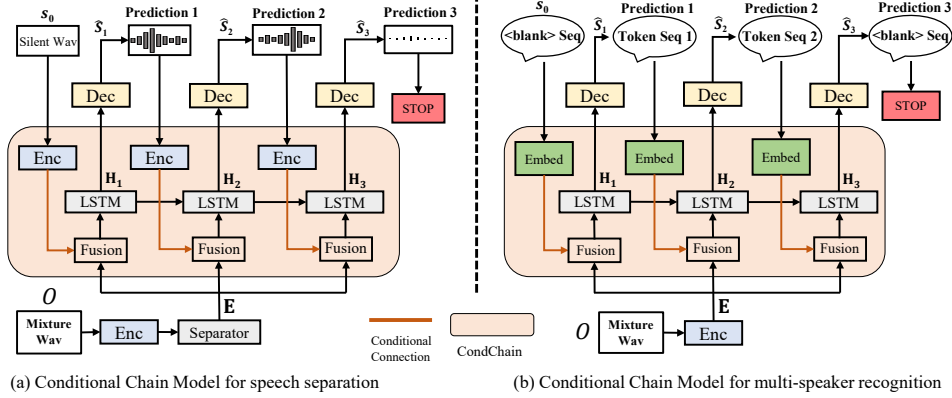

(a) Conditional Chain Model for speech separation  (b) Conditional Chain Model for multi-speaker recognition

Figure 2: Sequence-to-multi-sequence mapping with conditional model for multi speaker speech separation or recognition. In each sub-figure, the block with same name are all shared.

mapping for these two tasks. For both of them, the input sequence is from the speech domain. Let us first assume this to be an audio waveform $O \in \mathbb{R}^T$. Another common feature for these two tasks lies in that the output sequences are from the same domain ($\mathcal{D}_i = \mathcal{D}_j, i \neq j$), which means we could use a shared model at each step, i.e., $\mathrm{Encoder}_i$ in Eq. 2 and $\mathrm{Decoder}_i$ in Eq. 4 are respectively the same networks with different $i$.

For speech separation, the target sequences $s_i \in \mathbb{R}^T$ are all from the same domain as the input, i.e., $\mathcal{D}_i = \mathcal{O}$, and with the same length of the input mixture. Thus, we could use an identical basic Encoder (Enc in Figure 2(a)) to extract acoustic features from the input waveform $O$ and predicted waveform $\hat{s}_i$. As a contrast, multi-speaker speech recognition outputs a predicted token sequence $s_i \in \mathcal{V}^T$ with token vocabulary $\mathcal{V}$. We introduce an additional embedding layer, $\mathrm{Embed}$, to map these predicted tokens as continuous representations, which is used for conditional representation.

In speech separation, as illustrated in Figure 2(a), both the mixed audio and the predicted source will go through an Encoder ($\mathrm{Enc}$) to extract some basic auditory features. For the mixture waveform $O$, another separator ($\mathrm{Separator}$) will also be used as the function to learn some hidden representation which is suitable for separation. And both the $\mathrm{Enc}$ and the $\mathrm{Separator}$ form the process in Eq. 2. For the Fusion block, due to the same lengths from input and output, a simple concatenation operation is used to stack the feature dimension for each frame. For the $\mathrm{CondChain}$ in Eq. 3, we use a unidirectional LSTM. At each source step $i$, the Decoder ($\mathrm{Dec}$) is used to map the hidden state $\mathbf{H}_i$ into the final separated speech source. Multi-speaker ASR is also performed in a similar form, as illustrated in Figure 2(b). Note that we use connectionist temporal classification (CTC) [14] as a multi-speaker ASR network, since CTC is simple but yet powerful end-to-end ASR, and also the CTC output tokens without removing blank and repetition symbols can have the same length with the auditory feature sequence. Thus, we can realize the Fusion processing with a simple concatenation operation, similarly to speech separation.

## 3.2 Stop criterion

One benefit of the conventional seq2seq model is the ability to output a variable-length sequence by predicting the end of the sequence ($\langle EOS \rangle$ symbol) as a stop criterion. This advantage is inherited in our model to tackle the variable numbers of multiple sequences. For example, current speech separation or recognition models are heavily depending on a fixed number of speakers [21] or require extra clustering steps [15]. Thanks to the introduction of the above stop criterion, we can utilize the mixture data with various numbers of speakers during training, and can be applied to the case of unknown numbers of speakers during inference.

In our implementation, when we have the total number of output sequences as $N$, we attach an extra sequence to reach the stop condition during training. The target of this last sequence prediction for both speech separation and recognition tasks must be the silence, and we use the silent waveform and silent symbol (an entire $\langle blank \rangle$ label sequence in CTC), respectively.

For different tasks, the stop criterion should correspond to the form of target sequences. In speech separation task, we set the stop criterion as the prediction of silent waveform, implying that there is no more speech left. Similarly, in multi-speaker recognition, we encourage the last predicted utterance as all $\langle blank \rangle$ labels, which is a higher dimensional $\langle EOS \rangle$ used by seq2seq model. More explicitly, we use the average energy to determine pure silence in separation task and average posterior of <blank> label in ASR task to determine the end of prediction.

### 3.3 Training strategy with teacher-forcing and ordering

Like the conventional seq2seq approach [2], we use a popular teacher-forcing [47] technique by exploiting the ground-truth reference as a conditional source $s_{i-1}$. Teacher-forcing provides proper guidance and makes training more efficient, especially at the beginning of the training, when the model is not good enough to produce reasonable estimation. Considering the unordered nature of multiple sources in multi-speaker speech separation or recognition, we adopt a greedy search method to choose the appropriate permutation of the reference sequences. This method achieves good performance in practice while maintaining high efficiency. More details about teacher-forcing and reference permutation search could be found in Section B in the Supplementary Material.

## 4 Related Work

**Speech Separation** As the core part of the cocktail party problem [8], speech separation draws much attention recently [15, 17, 53, 21, 25, 22, 23, 26]. The common design of this task is to disentangle overlapped speech signals from a mixture speech with a fixed number of speakers, which is a typical example of the sequence-to-multi-sequence problem. Most existing approaches in this area follow the Parallel-mapping paradigm mentioned in Section 1, trained with permutation invariant training (PIT) technique [53]. This design should know the number of speakers in advance and could only tackle the data with the same number of speakers [36]. These constraints limit their application to real scenes, while our proposed structure can provide a solution to the variable and unknown speaker number issues. This study is inspired by recurrent selective attention networks (RSAN) [20], which has been proposed to tackle the above variable number of speakers in speech separation by iteratively subtracting a source spectrogram from a residual spectrogram. Similar ideas have also been proposed in time-domain speech separation [41] and speaker diarization [12]. However, the RSAN is based on the strong assumption of acoustic spectral subtraction in the time-frequency domain, and its application is quite limited. On the other hand, our conditional chain model reformulates it as a general sequence to multi-sequence transduction problem based on the probabilistic chain rule, which is applicable to the other problems including multi-speaker ASR than time-frequency domain speech separation. In addition, the relevance between current estimation and the former is learned by a conditional chain network in Eq. 3, which is more flexible and even applied to time-domain speech separation, making it totally end-to-end transduction from waveform to waveforms.

**Multi-speaker speech recognition** Multi-speaker speech recognition [46, 52, 33, 35], which aims to directly recognize the texts of each individual speaker from the mixture speech, has recently become a hot topic. Similar to the speech separation task, most of the previous methods follow the parallel mapping paradigm mentioned in Section 1. These methods could only tackle the data with the fixed number of speakers and require external speaker counting modules (e.g., speaker diarization [42, 1, 34]), which lose an end-to-end transduction function, unlike our proposed method.

## 5 Experiments

We tested the effectiveness of our framework with speech data as our primary testing ground, where the sequence mapping problem is quite common and important. To be specific, the following sections describe the performance of our conditional chain model towards multi-speaker speech separation and speech recognition tasks, compared to other baselines. Furthermore, we also evaluated a joint model of speech separation and recognition, using multiple conditions from both waveform and text domains. In the Section A of Supplementary Material, we provide the implementation details about all our experiments, and we also extend our model to one iterative speech denoising task in Section D.

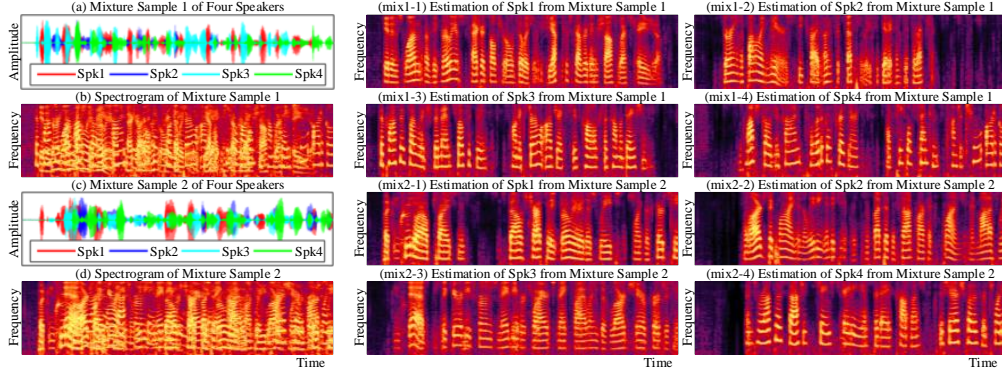

Figure 3: Visualization of two examples (mix1&mix2) with 4-speaker mixture from our WSJ0-4mix testset. For the mixture, both the waveform and spectrogram are showed. More examples and audios are available on our webpage: https://demotoshow.github.io/

Table 1: Performance for speech separation on WSJ0-mix dataset, compared with the same base model and SOTA methods. The $^*$ means the same base model with identical settings and hyper-parameters. The $N$ in WSJ0-$N$mix means the dataset with fixed $N$ speakers. All the "multiple architectures" based methods are trained specifically towards each specific $N$.

| Methods | Training Data | Training with variable number of speakers | Single architecture or multiple architectures | Eval (OC) SI-SNRi in WSJ0-Nmix | | | |
|---|---|---|---|---|---|---|---|
| | | | | 2mix | 3mix | 4mix | 5mix |
| RSAN [20] | WSJ0-2mix | × | single | 8.8 | - | - | - |
| OR-PIT [41] | WSJ0-2&3mix | ✓ | single | 14.8 | 12.6 | 10.2 | - |
| TasNet [23] | WSJ0-$N$mix | × | multiple | 14.6 | 11.6 | - | - |
| Our implemented TasNet$^*$ [50] | WSJ0-$N$mix | × | multiple | 15.4 | 12.8 | - | - |
| ConvTasNet [29] | WSJ0-$N$mix | × | multiple | 15.3 | 12.7 | 8.5 | 6.8 |
| Conditional TasNet$^*$ | WSJ0-2mix | × | single | 15.6 | - | - | - |
| | WSJ0-3mix | × | single | 12.7 | 13.3 | - | - |
| | WSJ0-2&3mix | ✓ | single | 16.3 | 13.4 | - | - |
| | WSJ0 2-5 mix | ✓ | single | 16.7 | 14.2 | 12.5 | **11.7** |
| DPRNN [29] | WSJ0-$N$mix | × | multiple | 18.8 | 14.7 | 10.4 | 8.4 |
| Voice Separation [29] | WSJ0-$N$mix | × | multiple | **20.1** | **16.9** | **12.9** | 10.6 |

## 5.1 Datasets

For the speech mixtures, i.e., the input $O$ for our tasks, with different numbers of speakers, data from the Wall Street Journal (WSJ) corpus is used. In the two-speaker scenario, we use the common benchmark called WSJ0-2mix dataset introduced in [15]. The 30 h training set and the 10 h validation set contains two-speaker mixtures generated by randomly selecting speakers and utterances from the WSJ0 training set *si_tr_s*, and mixing them at various signal-to-noise ratios (SNRs) uniformly chosen between 0 dB and 10 dB. The 5 h test set was similarly generated using utterances from 18 speakers from the WSJ0 validation set *si_dt_05* and evaluation set *si_et_05*. For three-speaker experiments, similar methods are adopted except the number of speakers is three. The WSJ0-2mix and WSJ0-3mix datasets have become the de-facto benchmarks for multi-speaker source separation, and we compare our results to alternative methods. Besides the separation task, we also instantiate our conditional chain model on multi-speaker speech recognition with the same WSJ0-2mix dataset. Although we just use the monaural speech in this work, it should be possible to apply our conditional chain model into the spatialized multi-channel speech, like the spatialized wsj0-mix, but we defer this to future work.

## 5.2 Multi-speaker speech separation

First, we investigate the application of our conditional chain model to speech separation benchmarks. We take TasNet as the main base model in Figure 2(a), which is a simple but powerful de-facto-standard method in speech separation. Table 1 reports the results with different training settings (the number of speakers), compared with the same base model. By following the convention of this benchmark, we use the downsampled 8 kHz WSJ0-2mix set to reduce the memory consumption

Table 2: The estimation counting results on WSJ0-mix test set with variable number of speakers (2 to 5 in our experiments). The overall accuracy of the counting is 94.8%. Here we set the threshold of the energy value per frame in the estimated speech as $3 \times 10^{-4}$ to judge whether to stop the iteration.

| Ref \ Est | 2 | 3 | 4 | 5 | 6 | Sum | Acc (%) |
|---|---|---|---|---|---|---|---|
| 2 | **2961** | 39 | 0 | 0 | 0 | 3000 | 98.7 |
| 3 | 12 | **2884** | 104 | 0 | 0 | 3000 | 96.1 |
| 4 | 0 | 42 | **2658** | 300 | 0 | 3000 | 88.6 |
| 5 | 0 | 0 | 125 | **2875** | 0 | 3000 | 95.2 |

of separation. We retrained the TasNet with the same settings and hyper-parameters from an open source implementation [50]. It should be noticed that the TasNet and most speech separation methods could only be trained and used in the same number of speakers, while our conditional chain model removes this limitation. In terms of the architecture of the model, we only added a single layer of LSTM compared with the base model, resulting in a negligible increase in the number of parameters.

The scale-invariant source-to-noise ratio improvement (SI-SNRi) results from Table 1 show that our conditional strategy achieves better performance than the base model (TasNet) under the same configuration. For the fixed number of speakers (2 or 3), our model improves on the original results, especially when there are more speakers (0.2 dB gains in WSJ0-2mix while 0.5 dB in WSJ0-3mix). Moreover, thanks to the chain strategy in our model, the WSJ0-2&3 mix datasets could be concurrently taken to train the model, and the performance is better than the training with each dataset. Compared with the RSAN [20] and OR-PIT [41] as mentioned in Section 4, which also could handle variable number of sources, our model achieves significant performance improvements as a result of our end-to-end speech transduction in time domain. To further verify the upper limit that our model can reach with more speakers in speech separation task, we remixed two datasets, which we refer to as WSJ0-4mix and WSJ0-5mix, by simply concatenating the given mixture list from the standard WSJ0-2&3mix. As we expect, without adding any additional speech sources, the performance trained with WSJ0-2to5mix gains further improvement in WSJ0-2&3 mix and get reasonable SI-SNRi results in even 4 and 5 speaker mixtures.

Besides the baseline models related to our methods, we also report the results from two strong works, i.e., DPRNN[26] and Voice Separation [29], which two upgrade the model architecture from the TasNet. Especially, Voice Separation [29] achieves the SOTA results in speech separation. However, their methods require *multiple* models (or multiple submodels in [29]) for each number of speakers in advance, which import additional model complexities or training procedures, and also cannot be applied to more speakers than the training setup. Even then, with the suboptimal base model (TasNet), our conditional chain model gets better results in 5 speaker mixtures. And we could clearly observe that as the number of speakers increases, our model achieves less performance degradation. We expect to realize further gains with our conditional chain model by improving the base models to DPRNN or Vocice Separation, which will be a part of our future work.

Furthermore, Table 2 reports the estimation counting accuracy with the trained WSJ0 2-5mix datasets. Here, we set the threshold of the energy value per frame in the estimated speech as $3 \times 10^{-4}$ to judge whether to stop the iteration, resulting in the overall accuracy of 94.8%, which is significantly better than the Voice Separation [29] ($\leq 62\%$). It is worth mentioning that the upper bound of speaker number was set as 5 in our trained model, so there is a strong tendency to predict silence at the 6-th step (similar with the observation from RSAN [20]), leading to higher accuracy for 5-speaker mixtures than the 4-speaker ones. We also visualize two examples from the WSJ0-4mix test set in Figure 3, where we observe clear spectrograms with the estimated sources from pretty tangled mixtures.

### 5.3  Multi-speaker speech recognition

Second, we evaluate our proposed method on the multi-speaker speech recognition task with the same WSJ0-mix corpus. We use the Transformer [43] as the basic Enc architecture block in Figure 2(b) to build speech recognition models, optimized with the CTC criterion [14]. Unlike the separation experiment in Section 5.2, which used 8 kHz, this section used the sampling rate as 16 kHz, which is the default setup for ASR to achieve better performance like most other previous works. An LSTM-

Table 3: WER (%) for multi-speaker speech recognition on WSJ0-$N$mix-**16 kHz** dataset.

| Methods | System | Training Data | WER | |
|---|---|---|---|---|
| | | | WSJ0-2mix | WSJ0-3mix |
| (1) DPCL + GMM-HMM [17] | HMM | WSJ0-2mix | 30.8% | - |
| (2) PIT-DNN-HMM [30] | HMM | WSJ0-2mix | 28.2% | - |
| (3) DPCL + DNN-HMM [28] | HMM | WSJ0-2mix | 16.5% | - |
| (4) PIT-RNN [6] | Attention-based | WSJ0-2mix | 25.4% | - |
| (5) PIT-Transformer | Attention-based | WSJ0-2mix | 17.7% | - |
| (6) PIT-Transformer | CTC | WSJ0-2mix | 31.2% | - |
| (7) Conditional-Chain-Transformer | CTC | WSJ0-2mix | 24.7% | - |
| (8) Conditional-Chain-Transformer | CTC | WSJ0-1&2&3mix | **14.8**% | 35.7% |

Table 4: SI-SNRi (dB) and WER (%) for multi-speaker joint training on WSJ0-mix-**8 kHz** dataset.

| Methods | Finetune Part | Condition | Training Data | WSJ0-2mix | |
|---|---|---|---|---|---|
| | | | | SI-SNRi | WER |
| Conditional TasNet + Pre-trained ASR | - | Wave | WSJ0-2mix | 15.2 dB | 25.1% |
| + With Multiple Condition | Separation | Wave + CTC | WSJ0-2mix | **15.5** dB | 17.2% |
| + With Joint Training | Separation, ASR | Wave | WSJ0-2mix | 12.0 dB | 15.3% |
| + With Multiple Condition | Separation, ASR | Wave + CTC | WSJ0-2mix | 10.3 dB | **14.4**% |

based word-level language model with shallow fusion is used during decoding [16]. We compare our conditional chain Transformer-based CTC with the other systems including the HMM systems (1): deep clustering (DPCL)+GMM-HMM [17], (2): PIT-DNN-HMM [30] , and (3): DPCL+DNN-HMM [28], and the attention-based systems (4): PIT-RNN [6] and (5): PIT-Transformer [3]. Note that all the PIT based methods correspond to the parallel mapping method in Figure 1(b), and they cannot deal with variable numbers of speakers.

We compare the effectiveness of the proposed conditional chain models with the same CTC architecture based on PIT in Table 3. Our conditional-chain Transformer (7) is significantly better than the corresponding PIT based system (6). Furthermore, the proposed conditional chain model can straightforwardly utilize the mixture speech data that has a variable number of speakers. To show this benefit, we train our model using the combination of the single and multi-speaker mixture WSJ0 training data. It can be seen that the conditional chain model trained with the combination of $1, 2$ and 3-speaker speech (8) achieves the best WERs, $14.9\%$, on the 2-speaker mixture evaluation set among all the other systems including (1) – (7). Also, the proposed method can be applied to the 3-speaker mixture evaluation set and achieves reasonable performance ($37.9\%$) in this very challenging condition. This result is consistent with the observation in the speech separation experiment in Section 5.2. Interestingly, by analyzing the hypothesis generation process, we found that the speaker with the longest text is predicted first generally, described in Section C in the Supplementary Material.

Our conditional models are trained in a greedy fashion. We experimentally prove that it is important in our method. One common way for the output order of the multi-output network is to use the pre-determined order of targets, which was also used in multi-speaker speech separation and recognition several years ago [11]. We compare the greedy method with pre-determined order in the speech recognition task using the (7)th experiment settings.. The greedy method and pre-determined order method achieve the WERs of 24.7% and 28.4%, respectively. The pre-determined order method is worse than the proposed greedy method.

## 5.4 Cross-domain condition in joint separation and recognition

In previous experiments of separation and multi-speaker speech recognition tasks, we explore the effectiveness of our conditional chain model with output sequences from the same domain, as depicted in Figure 2. In contrast, in this subsection, we evaluate a combination of cross-domain conditions by using the ASR output to guide separation learning. This direction is motivated by the so-called informational masking effect in the cocktail party problem [4, 19], where the linguistic clue may exert a strong influence on the outcome of speech-to-speech perception.

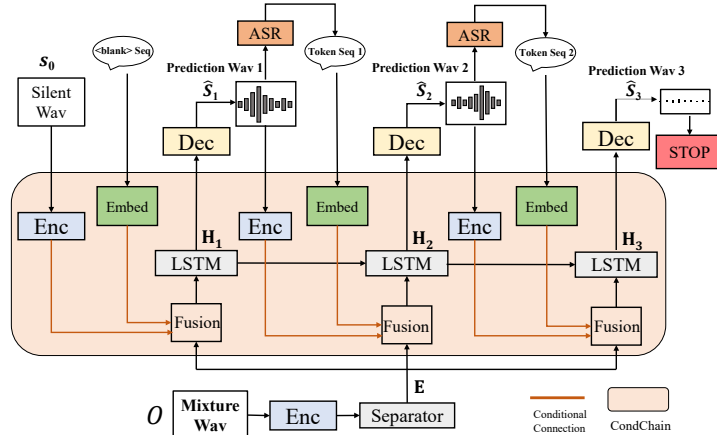

Figure 4: Conditional Chain Model for multi-speaker joint speech separation and recognition.

Specifically, as illustrated in Figure 4, in each step of our conditional chain model, both the separated waveform and the CTC alignment from the ASR model are utilized as the conditions for the next step, encouraging $\mathrm{CondChain}$ in Eq. 3 to jointly capture the information spanning multiple sources. The baseline conditional TasNet is trained using utterance-level waveform instead of chunk-level as in Section 5.2. In addition, unlike Section 5.3, we use a pre-trained ASR based on a single speaker Transformer-based CTC ASR system trained on the WSJ training set *SI284* without overlap with WSJ0-2mix test set to stabilize our joint model training. We also use the downsampled 8 kHz data for the reduction of the memory consumption in separation and non-truncation of the speech mixture for appropriate ASR evaluation. Details of framework are explained in the Appendix A.3.

The results are shown in Table 4. Note that the numbers listed in this experiment cannot be strictly compared with those in Sections 5.2 and 5.3 due to the above different configuration requirements between separation and recognition. When directly feeding the separated waveform from the baseline conditional TasNet to the pre-trained ASR, we get 25.1% WER. Finetuning of the TasNet with both waveform and CTC alignment conditions achieved 0.3dB improvement of SI-SNRi. The improvement shows that the semantic condition, such as CTC alignment, provides a good guide to the separation learning. Finetuning of both TasNet and ASR models with both waveform and CTC alignment conditions yields the best WER of 14.4%, while waveform-only conditioning obtains the WER of 15.3%. Note that this joint training severely degraded the SI-SNRi result, but the WER result gets a significant improvement. This intriguing phenomenon demonstrates that the separation evaluation metric (SI-SNRi) does not always give a good indication of ASR performance, as studied in [38, 39].

## 6 Conclusions

In this work, we introduced conditional chain model, a unified method to tackle a one-to-many sequence transduction problem for mixture signals. With the probabilistic chain rule, the standard sequence-to-sequence model is extended to output multiple sequences with explicit modeling of the relevance between multiple output sequences. Our experiments on speech separation and multi-speaker speech recognition show that our model led to consistent improvements with negligible increase in the model size compared with the conventional non-conditional methods,

In terms of the application scope, although we verify the proposed method with two specific tasks, speech separation and recognition, this conditional chain model can be flexibly extended to other sequence-to-multi-sequence problems for mixture signals, as a general machine learning framework. Therefore, as a future work, it might be interesting to adopt this method to other sequential tasks including natural language processing and video analysis. Another exciting direction would be introducing the attention mechanism into the fusion and conditional chain part, which could flexibly capture the implicit relationship between input and output from variable domains.

## Acknowledgement

This work was supported by the Major Project for New Generation of AI (Grant No. 2018AAA0100400), and the Strategic Priority Research Program of the Chinese Academy of Sciences (Grant No. XDB32070000).

## Broader impact

### Benefits

Our conditional chain model addresses the problems where one input sequence is mapped to multiple sequences by taking advantage of the intrinsic interaction between the output sequences. There are a variety of applications that can benefit from the use of the conditional information, such as the text generation tasks. Another important application is the cocktail party problem in speech processing. With the parallel mapping models, which are the dominant method at present, the model cannot handle the variable number of speakers flexibly due to the limitation of the model structure. In such models, the solution to label permutation problems is to exhaustively compute all the permutations with the computation cost of $N!$, which cannot be neglected when the number of speakers are more than 3. However, using the conditional model can avoid this problem. It also proves the effectiveness of our model which achieves relatively good performance in both separation and recognition tasks. We make a further step towards attacking cocktail party problem. This will improve the communication quality of human-computer interaction. And our method can also be applied in meeting transcription system to provide better performance. We would like to make our code available latter to facilitate the study applied to other tasks.

### Drawbacks

There is no doubt that the improvement of artificial intelligence can potentially revolutionise our societies in many ways. However, it also bring some risks to human's privacy. With the abusing use of speech separation and recognition techniques, hackers can easily monitor people's daily life, while a strong NLP system can also be applied to Internet fraud. We think the community should not only focus the development of techniques, but also concern the privacy issue. Besides, the widely use of artificial intelligence techniques may also lead to mass-scale unemployment problems, such as call center.

## Footnotes

[3] The model was proposed in [7] and was open source in ESPnet. We reproduced their results first and used the code to get the results on WSJ0-2mix data.

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
