[Supplementary Material]

# A  Implementation Details

## A.1  Speech separation experiments

For our conditional TasNet, we used the original configure from TasNet [19] with $N = 256, L = 20, B = 256, H = 512, P = 3, X = 8, R = 4$. More specifically, TasNet contains three parts: (1) a linear 1-D convolutional encoder that encapsulates the input mixture waveform into an adaptive 2-D front-end representation, (2) a separator that estimates a fixed number of masking matrices, and (3) a linear 1-D transposed convolutional decoder that converts the masked 2-D representations back to waveforms. We use the same encoder and decoder design as in [19], referring to the $\mathrm{Enc}$ and $\mathrm{Dec}$ in Figure 2(a) respectively. For separator, we set the channel number of the last $1 \times 1\ Conv$ as one, making it outputs single speech $\hat{s}_i$ at each step. And we move the last $1 \times 1\ Conv$ into the $\mathrm{Dec}$ part. For the Fusion and LSTM, $\mathbf{E} \in \mathbb{R}^{D^E \times T^E}$ is concatenated with the conditional state from the previous step $\mathrm{Enc}(\hat{s}_{i-1}) \in \mathbb{R}^{D^E \times T^E}$ at the feature's dimension. Then, a single layer of $\mathrm{LSTM}^{(2D^E \mapsto D^H)}$ is carried out to mapping the fused feature into $D^H$ dimension at each frame.

Also, we noticed the update of the base model could further improve the performance like the same tendency in [20, 22]. In this paper, we mainly focus on the relative performance over the original TasNet. For separation-related tasks, all the speeches are re-sampled to 8 kHz to make a fair comparison with other works.

For the training loss calculation, the negative SI-SNR metric is widely used in [46, 19, 20, 22, 25] and achieves satisfying performance. However, the SI-SNR will lead the predicted sources as a scaled signal compared with the ground-truth, making it totally mismatch in inference phase. To address this problem, we change the training loss from negative SI-SNR to negative SDR, forcing the prediction of speech signals as similar as possible with the ground-truth.

For the training strategy, we set the initial learning rate of $1 \times 10^{-3}$, which is multiplying by 0.9 every 8 epochs. In practice, we find the original ground-truth signals used as condition result in faster training speed, but a decrease in generalization ability. Therefore, to make the separation more robust, we add Gaussian noise with a standard deviation of 0.25 on these ground-truth source vectors (waveforms) during training.

## A.2  Multi-speaker speech recognition experiments

We basically use the open source package ESPnet [41] for the implementation of our ASR model. In the conditional Transformer-based CTC ASR model, there is a total of 16 Transformer layers, 8 before and 8 after the conditional chain LSTM. For the baseline Transformer-based CTC with PIT, there is a total of 12 Transformer layers in the acoustic model. The configuration of each Transformer layer is as follows: the dimension of attention is $d^{\mathrm{att}} = 256$, the dimension of feed-forward is $d^{\mathrm{ff}} = 2048$, number of heads is $d^{\mathrm{head}} = 4$. Before feeding the input to the Transformers, the log mel-filterbank features are encoded by two CNN blocks. The CNN layers have a kernel size of $3 \times 3$ and the number of feature maps is 64 in the first block and 128 in the second block.

## A.3  Multi-speaker joint speech separation and recognition

In the joint training experiments, we first pre-train two models for both separation and ASR tasks. The parameters of conditional TasNet are the same as our previous setup. For the pre-trained ASR, we train a single speaker Transformer-based CTC ASR model on the WSJ training set, which is a clean close-mic read speech corpus with about 80h. When jointly finetuning these two parts, we use the separated wave of TasNet as the input of ASR model and feedback the predicted CTC alignments of ASR into TasNet as an additional condition. Figure 4 shows an overview of our joint model.

Besides, we also introduce two extra teacher-forcing hyperparameters to control the optimization part of the joint model, which are $ss_{wav}$ and $ss_{ctc}$. The parameter $ss_{wav}$ is the probability of feeding separated wave to ASR, while $ss_{ctc}$ is the probability of inputting predicted CTC alignments as the condition. When aiming to optimize TasNet with multiple conditions, we fixed the parameters of ASR and set $ss_{wav} = 0$ and $ss_{ctc} = 1$, which means we only feed ground-truth wave to ASR and use the generated CTC alignments to guide the separation learning. When joint training both part to

Figure 4: Conditional Chain Model for multi-speaker joint speech separation and recognition.

improve the performance of ASR, we set $ss_{wav} = 0.5$ and $ss_{ctc} = 0.3$. All experiments only have access to the predicted wave and CTC alignments during inference.

# B   Training strategy with teacher-forcing and ordering

Figure 5: The procedure of selecting and sorting the target sequences using a greedy algorithm in our proposed methods. At each step $i$, the ground-truth sequence $s_i^*$ is selected from all the available references. In this example, the final order $\theta$ is $s_2 \rightarrow s_3 \rightarrow s_1 \rightarrow s_4$.

In Eq. 3, the neural network accepts the hidden state of the previous sequence that is estimated at the previous iteration. However, the estimation error at the previous iteration hurts the performance at the next iteration. To reduce the error, we use the teacher-forcing [43] technique, which boosts the performance by exploiting ground-truth reference. During training, Eq. 3 is replaced with as follows:

$$\mathbf{H}_i = \text{CondChain}(\mathbf{E}_i, \mathbf{s}_{i-1}^*), \tag{5}$$

Here, $s_{i-1}^*$ is a ground-truth sequence of index $i-1$. The teach-forcing technique is commonly used in conventional seq2seq methods. However, target sequence in seq2seq is determined and has an immutable order, so the previous approaches also generate the sequence through a fixed order, either from the beginning to the end, or the reverse order. But for many seq2MSeq problems, the multiple reference sequences are unordered. There arises a problem about how to select the $s_i^*$ from the $\mathbf{S}$ to process the next iteration, which is also how to determine the best order $\theta$ of target sequences.

One most straightforward method is to use the permutation invariant training (PIT) strategy to traverse all the permutations and select the optimal one to update the parameters. However, with the teacher-forcing technique attending each step of the output, we must go through the whole feedforward process for each permutation, which takes too much computational complexity. To alleviate this

problem, we examine one simple greedy search strategy. As shown in Figure 5, for each output iteration $i$, the optimal target index is selected by minimizing the difference (distance) with $\hat{s}_i$ among a set of the remaining target set, and the selected ground-truth sequence $s_i^*$ is fed into the next decoding iteration. With this greedy strategy, the repetitive computation only occurs at the calculation of distance, and there is no need to re-run the feedforward process. In addition, the total number of repetitive computation is $\sum i = N(N+1)/2$, compared with the $N!$ in PIT based strategy.

## C  Analysis on Speech Recognition Outputs

Our method is trained in a greedy fashion, which does not address the label permutation problem from the formulation as other methods do, such as permutation invariant training (PIT) or deep clustering (DPCL), etc. We further look into the speech recognition results in terms of the order following which our model generates the hypotheses. In Table 5, we show the confusion matrix of the prediction order and the text length ranking for the two-speaker scenario. We observe that the order is somehow correlated to the length of the text. In $88\%$ of evaluation samples, the generation order is consistent with the length ranking. By considering two texts sequences that may have very close lengths or some words are much simpler than the others, we loosen the constraint for the length ranking. If we simply accept those cases where the reference text of the first hypothesis is shorter than the second, but within a range, the pattern is more obvious. For example, the result is shown in Table 6 when the range is 5. In $99\%$ evaluation samples, the generation order is consistent with the length ranking. In the three-speaker scenario, we find a similar pattern: the first hypothesis is significantly longer than the other two, $98\%$ when the range is 5. We also have the same conclusion on the Transformer-based CTC ASR system trained with PIT in two speaker scenario. It is a Parallel-mapping framework without such conditional dependency. However, we found that the output of each head is highly dependent on the lengths. In our model, $87\%$ of output from one head is longer than that from the other. If we further consider the range of 5, the ratio becomes $99\%$. Perhaps this can be the heuristic information to address the label permutation problem.

Table 5: Confusion matrix between the hypothesis (Hyp.) generation order and the order of reference (Ref.) text length in 2-speaker case.

| Hyp. \ Ref. | long | short |
|---|---|---|
| 1st output | 2627 | 373 |
| 2nd output | 373 | 2627 |

Table 6: Confusion matrix between the hypothesis (Hyp.) generation order and the order of reference (Ref.) text length in 2-speaker case with loosing range 5.

| Hyp. \ Ref. | long | short |
|---|---|---|
| 1st output | 2965 | 35 |
| 2nd output | 35 | 2965 |

## D  Implementation for iterative speech denoising

Former experiments on multi-speaker speech separation and recognition show the effectiveness of our conditional chain to disentangle the input mixture signals into several components. For our proposed tasks above, actually, there is a mutually exclusive relationship between the outputs. However, the seq2Mseq tasks also cover some instances that the output sequences get a positive correlation. In this section, we manage to verify the ability to model this positive correlation, besides the proposed tasks shown in Section 5.

In the speech domain, the problem of a positive correlation between multiple outputs is also reflected in some problems. In this section, we take the speech denoising task as an example to verify the effectiveness of our conditional chain model in the case of positive correlation between iterative steps. The iterative estimation of some signals is an effective technique in speech processing, which could be used in speech enhancement [10], i-vector estimation [23], speech separation [17]. Similar to this technique for speech denoising, we implement our conditional chain model with two iteration in the chain to denoise the noisy input speech. This formulation is very similar to the iterative re-estimation of the clean signal. That is to say that our conditional chain method is trained with two identical references as objects, i.e., $s_1 = s_2 = s$. And the output of the second step is conditioned on the estimation from the first step, similar to the structure shown in Figure 2(a).

To evaluate this, we conduct the speech denoising task based on a recently published dataset from the DNS-Challenge 2020 [27], which consists of 60,000 no-reverberant noisy clips in training and 300 in evaluation set.

Table 7: The SDR performance on no-reverberant testset in DNS-challenge 2020.

| Methods | SDR | SDRi |
|---|---|---|
| Official baseline[45] | 13.4 | 4.2 |
| Our implementation for [45] | 14.6 | 5.5 |
| TasNet | 17.3 | 8.2 |
| Conditional TasNet (1st step) | 17.8 | 8.7 |
| Conditional TasNet (2nd step) | **18.0** | **8.9** |

Here, we compare the results with the official baseline model [45] and our implemented baseline based on TasNet. And, we also report the performance of our conditional chain model with the same architecture and hyper-parameters with the base TasNet model.

From the results in Table 7, we could see that, with the iterative estimation of the clean speech signal in our conditional chain model, the performance gets obvious improvement over the same base model (TasNet). And, the estimation of the second step is better than the first step. These results show that our conditional chain learns to refine the condition from former steps, which further proves that our model has good adaptability and generalization performance when learning the relationship between multiple output sequences.