[Reviews · NeurIPS 2020]

Review 1

Summary and Contributions: (After reading the author feedback: Thanks for extending the results section with the suggested experiments. They addressed my concerns about the experiments, hence, I increased my score accordingly). This paper presents a framework for sequence to multi-sequence generation where each of the generated sequences is conditioned on previously generated sequences as well as then input sequence. Experiments are conducted on two datasets of cocktail party speech; one output individual audio streams, and the other produces speech recognition output of each stream.

Strengths: 1) The paper presents a general framework for seq2multi-seq transduction with good experimental results on two speech datasets. 2) The use of a stopping criterion allow the model to decide on the total number of output streams which can be variable.

Weaknesses: 1) the two used datasets are very related, where the input sequence is cocktail party speech, with one outputting the audio of each stream and the other producing the ASR output of each stream. More experiments with sequences of different characteristics which interact in more complicated ways will show the value of the proposed method, e.g. speech translation (audio --> ASR and MT), (audio --> ASR and audio event). 2) The value of the multi-sequence ordering is not emphasized. What is the final output if a random ordering is used? 3) The used data is synthetic. Do you have results for cases of someone talking over another in natural dialogues, e..g meeting datasets, or conversational telephone speech?

Correctness: probably correct

Clarity: Clear enough

Relation to Prior Work: The paper need to show empirical results against other baseline systems: https://www.aclweb.org/anthology/P18-1244.pdf https://arxiv.org/pdf/2002.03921.pdf

Reproducibility: Yes

Additional Feedback: 1) In equation (1) serial mapping can still be conditioned on input sequence O, i.e. p(S|O) = p(s1|O) PROD_{i=2}^{N} p(s_i | s_{i-1}, O}. This way serial mapping can consume encoder representation of O as well as one previous output sequence. 2) The proposed method factorizes the joint distribution into: p(S|O) = PROD_{i=1}^{N} p(s_i |s1, .., s_{i-1}, O}. The conditioning over the whole history of sequences is manifested in the model through passing the LSTM hidden layer between sequences in the decoder. The serial mapping I mentioned in the point above can be tested by using a random/zero initial LSTM hidden vector for each sequence in the decoder (no lateral LSTM arrow in figure 2). It it is important to show the value of such conditioning as it is central to the proposed approach.


Review 2

Summary and Contributions: I thank the authors for their response to review comments. The additional clarifications, experiments in reverberant conditions, and on other tasks clearly improves the submission. I have increased my score accordingly. ------------------------------------------------------------------------ The authors propose an architecture for learning multiple output sequences from a single input sequence. Typical approaches for generating multiple output sequences involve multitask learning or chained modeling. But the proposed approach conditions on both inputs and prior outputs of the model. The proposed approach is applied on multitalker speech separation and multitalker automatic speech recognition tasks. The application of this approach to these tasks is interesting and novel compared to other previously proposed approaches. Arguing that the output domains of the multiple sequences are identical, the authors use the same model with shared parameters, sequentially estimating targets, conditioning on the previous outputs. Results on widely used benchmark datasets for these tasks show improvement compared to previous approaches and better generalization as the number of output sequences (number of speakers in a mixture) increases.

Strengths: The main strength of the paper, in my opinion, is in proposing an architecture that improves multitalker speech separation and ASR tasks. The proposal can be used to separate N speakers without having to train a single system for each N, which is the typical strategy to handle multiple speakers. The proposed architecture, which I think is quite clever, does this by sharing model parameters, but varying the inputs that are passed to the model at various stages. The results are also quite strong; the authors show that the method beats previous techniques that specifically train separate systems for different values of N. The claim that the proposed architecture is general enough to be used for arbitrary multi-output sequence tasks is not substantiated (see weaknesses).

Weaknesses: Generality: The idea is not as general as has been presented. It is quite similar to techniques like multisource decoding [1], where the decoder of a network is conditioned on multiple input sequences, and deliberation models [2]. Furthermore, if one uses attention based models that directly model p(Y|X), conditioning multiple sequences is quite straightforward, and not uncommon. For example, if there are multiple sequences Y1, Y1, by concatenating Y1 and Y2, one can model p(Y1, Y2 | X) = P(Y1 | X) P(Y2 | Y1, X), which naturally conditions on both input and prior sequences (see, e.g., [4]). The novelty lies mostly in how this is being applied to multitalker separation tasks, or specifically, multiple tasks from the same domain where order of the output doesn’t matter much. Multiple domains, variable output sequence lengths: Both tasks that the authors chose share output domains and have the same lengths as the input. If the output sequences come from different domains, for example, speech to text and NLP-tokens, it’s unclear how the proposal will be different from existing literature (e.g., [4]), since the authors will most likely have to define an order and most likely cannot share model parameters. Conditioning the way the authors are proposing (Fusion, in the paper) also becomes more complex; and using attention would make the model very similar to ideas proposed in [1] and [2]. All the results are in non-reverberant conditions. Although separating 4-5 speakers are interesting and hard tasks, perhaps they are less realistic than separating 2-3 speakers in the presence of reverberation and other noise sources. It would be interesting to see how the methods work in the presence of reverberation (e.g., using spatialized wsj0-mix [5]). References [1] Zoph, Barret, and Kevin Knight. "Multi-source neural translation." arXiv preprint arXiv:1601.00710 (2016). [2] Xia, Yingce, Fei Tian, Lijun Wu, Jianxin Lin, Tao Qin, Nenghai Yu, and Tie-Yan Liu. "Deliberation networks: Sequence generation beyond one-pass decoding." In Advances in Neural Information Processing Systems, pp. 1784-1794. 2017. [3] Hu, Ke, Tara N. Sainath, Ruoming Pang, and Rohit Prabhavalkar. "Deliberation Model Based Two-Pass End-to-End Speech Recognition." In ICASSP 2020-2020 IEEE International Conference on Acoustics, Speech and Signal Processing (ICASSP), pp. 7799-7803. IEEE, 2020. [4] Haghani, Parisa, Arun Narayanan, Michiel Bacchiani, Galen Chuang, Neeraj Gaur, Pedro Moreno, Rohit Prabhavalkar, Zhongdi Qu, and Austin Waters. "From audio to semantics: Approaches to end-to-end spoken language understanding." In 2018 IEEE Spoken Language Technology Workshop (SLT), pp. 720-726. IEEE, 2018. [5] https://www.merl.com/demos/deep-clustering

Correctness: Presented techniques are correct and sound, although the claim of the proposed approach being general enough for any multiple output sequence task is not substantiated. Results are for a very specific domain for which the presented ideas are particularly well suited.

Clarity: Mostly well written, and easy to understand. English usage can be improved in several places (see detailed comments). TasNet citation in supplementary material points to the PIT paper, when the authors most likely meant Conv-TasNet work. And in the paper, baseline TasNet most likely means Conv-TasNet, looking at the results and citation.

Relation to Prior Work: Prior work mostly addresses speech separation and ASR, and not multitarget modeling in general, e.g., in NLP and machine translation as has been pointed out earlier. Consider discussing those as well.

Reproducibility: Yes

Additional Feedback: Minor comments / typos: Abstract: relevance between -> relationship between While discussing the idea in Introduction and Sec. 2, it’s not obvious why the proposal doesn’t order outputs; it becomes clear only when the authors discuss its application in multitalker separation. Consider making that clear early on. Sec. 2: “For our conditional chain mapping, … relevance from data, even if it seems very independent intuitively.”: Unclear what the authors mean. Consider rephrasing. Sec. 3.1, second paragraph: Description about identical base encoder sounds repetitive since this was already discussed in the previous paragraph. Sec. 3.2: For the stop criterion, how do the authors determine that the predicted sequence is all-silence or all-blank? Does every single frame have to be pure silence (energy less than threshold) or blank (argmax of posterior is indeed blank), or is it based on some threshold (e.g., 90% of the frames)? Sec. 3.2: “when we have a total number of output sequences as N,..” Description is a little confusing. Consider rephrasing. Perhaps it is enough to say that a blank / silence sequence is always appended to help predict the end of sequences. Sec 4: speech separation draws much attention -> speech separation has drawn a lot of attention Sec. 4: acoustic spectral subtraction in the time-frequency domain, and it’s application is quite limited: Unclear what the authors mean here. Is separation in the time-frequency domain limiting? Why? Sec. 5.1: is used by us -> is used. Sec. 5.1: introduced by -> introduced in Fig. 3: Please also include ground truths. It is not clear if separation is doing a good job otherwise. Tab. 1: Are WSJ 4Mix and WSJ 5Mix results on the same sets? Is this included as part of the released corpus? Sec. 5.2: “It should be noticed that TasNet and most…” Repetitions of what has been described before. Consider removing it. Sec. 5.2: Even though -> Even then Sec. 5.2: WSJ-5Mix, by simply concatenating: Are the utterances generated by concatenation without any overlap? Please clarify. Sec. 5.4 second paragraph: Overall, the description is a little hard to follow. Please consider rewriting it. For example, it is not clear whether the separated speech and CTC alignments match (only the supplementary material makes that clear). Unclear what the authors mean by utterance-level vs. chunk-level. What is the training instability that mixing addresses? Tab. 2: Perhaps it’s easier to include % instead of the actual counts. Tab. 3: What is the WER on the standard WSJ test set (1 speaker) using the proposed system. Tab 3. Why is there a big jump in WER when including 1, 2, 3Mix utterances into the training corpora? Would the baselines benefit from similar changes to the training data? Number sections about broader impact / drawbacks like the rest. Drawbacks section: “human’s daily life” -> “people’s daily life”.


Review 3

Summary and Contributions: This paper proposes an elegant extension to seq2seq models to a conditional multi-sequence model. These models are able to infer multiple output sequences by conditioning on both input and previously estimated output sequences. The paper shows the use case of these models on multi-speaker speech recognition and speech separation tasks. >>>> I thank the authors for their detailed response to the comments. The clarifications and experiments certainly improve the submission quality. I also acknowledge that I had missed some of the weaknesses that other reviewers mentioned. So overall I would like to keep my scores unchanged. <<<<

Strengths: 1) The paper proposes a conditional chain model for sequence to multi-sequence problem. Here the model output is conditioned on all previous sequence and the encoder representation. The authors have elegantly used an lstm to introduce this conditional dependence. 2) The authors have also processed this conditional chain in a seq2seq manner. In the sense that the model is not dependent on fixed number of output sequences. This allows the model to be flexible and be trained on all kinds of speech data (single speaker, multispeaker .. ) 3) Although the models use a CTC based loss they are able to outperform attention based CE models.

Weaknesses: 1) It seems like the model really works well when there is a mixture of datasets i.e. single, dual and multi speaker. Would be interesting to see dependency on this? 2) It seems like the model is limited to CTC loss, would it be possible to train them towards attention based enc-dec training?

Correctness: 1) The authors have provided code based out of an open-source project. So the work seems easily reproducible.

Clarity: The paper is clearly written with a supported website showing examples of how this model works.

Relation to Prior Work: Mostly adequate. However I feel there is some missing related work with adding context to ASR like - 1) Deep context: end-to-end contextual speech recognition https://ieeexplore.ieee.org/stamp/stamp.jsp?arnumber=8639034 2) Gated Embeddings in End-to-End Speech Recognition for Conversational-Context Fusion https://arxiv.org/pdf/1906.11604.pdf

Reproducibility: Yes

Additional Feedback: 1) Have you tried this approach on the CHiME-5/6 dataset? It seems like the model could do really work in a cocktail-party problem. 2) Would the model work for far-field and close-field speech separation? Or the speakers should really be at the same distance from the microphone. 3) Is the CTC model rescored with a language model? 4) How is the model stability during training? As the conditional representation would really take some time to be reasonable.


Review 4

Summary and Contributions: This paper presents a novel approach addressing the problem of sequence to multi-sequence learning named Conditional Chain Mapping. In the proposed approach, each output sequence is inferred based on the input sequence and the previously inferred output sequences. The approach is applied to two speech-related tasks: speech separation and speaker recognition. Experiments are presented on both tasks, where the proposed approach is shown to yield similar performance to strong baselines from the literature.

Strengths: The conditional chain mapping approach is novel as far as I can tell, and it's very interesting. It could be beneficial for different tasks, hence it is a significant contribution.

Weaknesses: The experiments demonstrates the viability of the approach, however I have a few concerns: - In Table 1, it seems to me that some comparisons are unfair because of the different training sets. For instance the Conditional TasNet trained on WSJ0 2-5 Mix seems to be trained on more data than DPRNN for 2mix, the authors should clarify that. - The authors selected TasNet for their base model, but it seems that DPRNN and Voice Separation are better, why not applying the Conditional Chain Mapping to them? - Please explain briefly the SI-SNR metric. - On Table 4, experiment (8) seems unfair to use as comparison as it has more training data. Also, I don't see the point of reporting WSJ0-3mix if only one model can do it.

Correctness: It is correct as far as I can tell.

Clarity: The paper is clearly written. Section 5 is sometimes not easy to follow due to the placement of the tables.

Relation to Prior Work: The related work about the speech tasks used in the paper (Speech Separation and Multi-speaker speech recognition) is clearly discussed, however a brief discussion on sequence-to-sequence mapping is missing from Section 4.

Reproducibility: Yes

Additional Feedback: Overall this paper is novel and is a significant contribution, however the experiments are not clear. I would be happy to increase my score if the authors address my concerns. It would also have been nice to see another application in another domain, like NLP for instance. Edit after rebuttal: I've read the rebuttal. The authors clarified the experiments, hence I've increased my score.

[Author Response · NeurIPS 2020]

We thank all the reviewers for your careful reading of this paper and constructive comments. We did our best to answer all of the questions and supplement the experiments as much as possible in this limited rebuttal period.

**For Reviewer 1**: **(3.1)** We conducted speech translation experiments on the Fisher and CALLHOME Spanish-English corpus with the proposed conditional model. Although we could not finish the entire training and tuning hyper parameters due to the limited time, we obtained promising results on a validation set at the 12th training epoch (BLEU: 67.2), compared with the Transformer model (BLEU: 63.1), with the same configuration. We will add the experiments to Section 5 to provide more evidence about the generalization of our work. **(3.2)** As you suggested, we trained a conditional ASR model with a similar configuration for Method 7 in Table 3, except that the labels and conditions are randomly ordered. The WER of random ordering is $28.4\%$, worse than the greedy method $24.7\%$. **(3.3)** We fully agree with this point and we will apply our method to more realistic scenarios, e.g. CHiME6 and CALLHOME, as the most important future work. We are confident that our method can be used in such scenarios, since the related study of our conditional model in speaker diarization actually [10] got very positive results in the CALLHOME task. **(6)** The two papers you mentioned use a different dataset (based on **WSJ1**, which is *five-times larger* than **WSJ0**) and we cannot directly compare them. Instead, we performed the experiment with the exact same WSJ0-2mix data by using our reproduced version of [6] (Method 5 in Table 3), which is similar to the first work you mentioned. We will clarify this point in the manuscript. **(8)** As you suggested, we trained our conditional model with zero initial LSTM hidden vector for each sequence in the decoder with the same configuration as Method 7 in Table 3. The WER on WSJ0-2mix was degraded by $8.6\%$, which proves that the whole history information does help. This result will be added to the paper.

**For Reviewer 2**: **(3)** We really appreciate your comments about the relationship with several related studies. We fully agree with your point and we did not intend to claim that the conditional methodology itself is novel. Instead, we wanted to claim that the novelty of this paper is to apply such a concept to analyze "Mixture Signals" (as shown in the title and the other part of the paper) especially when the number of sources is not known. This often happens in audio/speech signals, especially for multispeaker scenarios, as you pointed out. We tried to clarify this point during the submission stage, but we will further clarify this point in our abstract and introduction. In addition, we re-trained our model in the spatialized wsj0-2mix dataset, compared with the same configuration and network from an open-source implementation with 8-ch MVDR beamformer in the time-frequency domain. Our results on the 2-mix reverberant condition get similar results with the baseline (9.57db vs 9.78 db SDR) without much tuning, which proves the adaptability of our method even under reverberant conditions. We will continue these experiments and add the results into our appendix part. **(8)** Thanks a lot. We will polish the article with your suggestion. (Sec 3.2) For stop criterion, we use average energy to determine pure silence in separation task (see line 241) and average posterior of <blank> label in ASR task. (Tab. 3) We agree with your point and applied the 1,2,3Mix utterances into the training corpora for baseline models. As you pointed out, this additional experiment shows the similar performance improvement ($19.2\%$ for 2mix and $39.0\%$ for 3mix), but the proposed model is still better ($14.9\%$ for 2mix and $37.9\%$ for 3mix), which will be added to Table 3.

**For Reviewer 3**: **(3.1)** Because of the chain design of our model, we could make better use of different datasets (e.g. with a different number of speakers). Experimentally, our model does improve from the mixing of multiple data sets, but we also achieve better results for a single data set than the baseline approach (see in Table 1&2). This proves the effectiveness to use the conditional chain to fit the relationship between multiple outputs even for a single dataset. **(3.2)** Yes, one of our future work is to introduce the attention mechanism into the fusion and conditional chain part, which could flexibly capture the implicit relationship between input and output from variable domains, as we mentioned it in Section 6. Actually, we already applied it to speech translation tasks, and it seems to be working (See Reviewer 1(3.1)). **(8.1)** and **(8.2)** Currently, our method is only applied to the near-field condition, but some extended experiments (e.g. spatialized WSJ0-mix, see in reply for Reviews 2 (3)) have been done and showing promising results under the reverberant, far-field, and various mic/source location conditions. Our ongoing work is to apply it to more realistic CHIME-5/6 datasets, as you suggested. **(8.3)** Yes, a word-LM is used. This is described in Section 5.3 first paragraph, and we will rephrase this paragraph to clarify it. **(8.4)** In all our experiments, the model seems to be quite stable. We did experiments in several different environments, and the problem of repeatability did not arise.

**For Reviewer 4**: **(3.1)** We appreciate your comments about the lack of clarification about the different training sets. We agree that Table 1 is misleading and we'll add more detailed information about the training data. Also, we are working on the fair comparisons with DPRNN (see the discussion below). **(3.2)** We agree with your point. Among Voice Separation and DPRNN, we actually picked up DPRNN in addition to TasNet, and performed experiments based on the conditional-DPRNN model during the submission stage. The reason we did not list the DPRNN result is that our implemented DPRNN could not reproduce comparable results on WSJ0-2mix (18.0 vs 18.8 dB SI-SNRi in [24]). However, the results with our implementation also shows a similar tendency (e.g., the conditional-DPRNN is 0.2 dB better than the DPRNN in WSJ0-2mix and the WSJ0-2&3 training gets further improvements of 0.3 dB. We can add our DPRNN result to the paper once we reproduce the comparable numbers in [24]. **(3.3)** We will add the explanation of SI-SNR accordingly. **(3.4)** Thanks for pointing out this. We conducted experiment on the baseline model with 1, 2, 3Mix utterances. Please check our reply for Review 2 (8, Tab.3).

[Meta-Review · NeurIPS 2020]

All reviewers agree that the paper is an interesting contribution (a conditional chain model), for an important problem (multi-sequence problem, with application to ASR). There was concerns about the experimental section on the weak side, as well as some unclear points. However, reviewers found the rebuttal convincing enough and raised their scores accordingly.